# Multi-Epitopic Peptide Vaccine Against Newcastle Disease Virus: Molecular Dynamics Simulation and Experimental Validation

**DOI:** 10.3390/vaccines12111250

**Published:** 2024-11-01

**Authors:** Muhammad Tariq Zeb, Elise Dumont, Muhammad Tahir Khan, Aroosa Shehzadi, Irshad Ahmad

**Affiliations:** 1Department of Molecular Biology and Genetics, Institute of Basic Medical Sciences, Khyber Medical University, Phase-V, Hayatabad Peshawar, Peshawar 25100, Pakistan; drtariqzeb@gmail.com; 2Genomic Laboratory, Veterinary Research Institute, Bacha Khan Chowk, Charsadda Road, Peshawar 25100, Pakistan; 3Institut de Chimie de Nice, Université Côte d’Azur, CNRS, UMR 7272, 06108 Nice, France; elise.dumont@univ-cotedazur.fr; 4Institut Universitaire de France, 5 Rue Descartes, 75005 Paris, France; 5Institute of Molecular Biology & Biotechnology (IMBB), The University of Lahore, KM Defence Road, Lahore 54000, Pakistan; aroosa.shahzadi4242i@gmail.com; 6State Key Laboratory of Respiratory Disease, Guangzhou Key Laboratory of Tuberculosis Research, Department of Clinical Laboratory, Guangzhou Chest Hospital, Institute of Tuberculosis, Guangzhou Medical University, Guangzhou 510180, China; 7Qihe Laboratory, Qishui Guang East, Qibin District, Hebi 458030, China

**Keywords:** Newcastle disease virus, synthetic peptide vaccine, immunogenicity, epitope, molecular dynamics

## Abstract

Background: Newcastle disease virus (NDV) is a highly contagious and economically devastating pathogen affecting poultry worldwide, leading to significant losses in the poultry industry. Despite existing vaccines, outbreaks continue to occur, highlighting the need for more effective vaccination strategies. Developing a multi-epitopic peptide vaccine offers a promising approach to enhance protection against NDV. Objectives: Here, we aimed to design and evaluate a multi-epitopic vaccine against NDV using molecular dynamics (MD) simulation. Methodology: We retrieved NDV sequences for the fusion (F) protein and hemagglutinin–neuraminidase (HN) protein. Subsequently, B-cell and T-cell epitopes were predicted. The top potential epitopes were utilized to design the vaccine construct, which was subsequently docked against chicken TLR4 and MHC1 receptors to assess the immunological response. The resulting docked complex underwent a 1 microsecond (1000 ns) MD simulation. For experimental evaluation, the vaccine’s efficacy was assessed in mice and chickens using a controlled study design, where animals were randomly divided into groups receiving either a local ND vaccine or the peptide vaccine or a control treatment. Results: The 40 amino acid peptide vaccine demonstrated strong binding affinity and stability within the TLR4 and MHC1 receptor–peptide complexes. The root mean square deviation of peptide vaccine and TLR4 receptor showed rapid stabilization after an initial repositioning. The root mean square fluctuation revealed relatively low fluctuations (below 3 Å) for the TLR4 receptor, while the peptide exhibited higher fluctuations. The overall binding energy of the peptide vaccine with TLR4 and MHC1 receptors amounted to −15.7 kcal·mol^−1^ and −36.8 kcal·mol^−1^, respectively. For experimental evaluations in mice and chicken, the peptide vaccine was synthesized using services of GeneScript Biotech^®^ (Singapore) PTE Limited. Experimental evaluations showed a significant immune response in both mice and chickens, with the vaccine eliciting robust antibody production, as evidenced by increasing HI titers over time. Statistical analysis was performed using an independent *t*-test with Type-II error to compare the groups, calculating the *p*-values to determine the significance of the immune response between different groups. Conclusions: Multi-epitopic peptide vaccine has demonstrated a good immunological response in natural hosts.

## 1. Introduction

Newcastle disease (ND), caused by avian paramyxovirus type 1 (APMV-1) [1], is a highly contagious viral infection that affects birds, particularly poultry. ND primarily affects domestic poultry, including chickens and turkeys, but it can also infect a wide range of wild bird species [2]. Clinical signs of ND vary widely depending on the virulence of the virus strain and the species of bird affected. These signs can include respiratory distress (coughing, gasping, and nasal discharge), neurological symptoms (tremors, twisted necks, and paralysis), gastrointestinal issues (diarrhea), and a drop in egg production or abnormal eggs in laying birds. In severe cases, ND can cause sudden death with high mortality rates. ND has a significant economic impact on the poultry industry due to mortality, reduced egg production, trade restrictions, and the cost of control measures [3,4].

Newcastle disease virus (NDV) is a remarkable member of the *Paramyxoviridae* family, specifically classified as avian paramyxovirus type 1 (APMV-1). Its genome is a single-stranded, negative-sense RNA molecule [5]. The genome is approximately 15,000 nucleotides in length and encodes six major proteins: nucleoprotein (NP), phosphoprotein (P), matrix protein (M), fusion protein (F), hemagglutinin–neuraminidase protein (HN or H), and the large polymerase protein (L) [6,7]. The NP and P proteins encapsulate the viral RNA [8] to form the ribonucleoprotein (RNP) complex, which serves as a template for viral RNA synthesis [9]. The M protein plays a crucial role in virus assembly and budding [6]. The F and HN proteins are responsible for viral entry into host cells, facilitating membrane fusion and attachment, respectively [10]. The L protein acts as the viral polymerase, directing the transcription and replication of the viral genome [1]. NDV’s genome also exhibits genetic diversity, with various strains ranging from low pathogenicity to high pathogenicity, impacting the severity of disease in birds [11]. Understanding the intricacies of NDV’s genome is essential for developing effective vaccines and antiviral strategies to mitigate its impact on poultry populations.

Vaccination is a key method for preventing ND in poultry. Different vaccines are available to protect against various virus strains [12,13,14]. The F and HN play crucial roles in developing Newcastle disease virus (NDV) vaccines [15,16,17]. These two viral proteins are potential targets for vaccine development because they are involved in key aspects of the virus’s life cycle and immune response. The HN protein is also immunogenic and can stimulate an immune response. Antibodies against the HN protein may prevent the initial attachment of the virus to host cells, reducing infection [18,19,20,21]. In developing NDV vaccines, the fusion protein (F) and hemagglutinin–neuraminidase protein (HN) play pivotal roles. The F protein is instrumental in viral entry, mediating the fusion of the virus with host cell membranes, making it a potent immunogen capable of inducing neutralizing antibodies.

On the other hand, the HN protein has a dual function, aiding in viral attachment to host cells through receptor binding and facilitating viral release by neuraminidase activity. Both proteins are highly immunogenic and contribute to the overall efficacy of NDV vaccines. Vaccines can stimulate robust immune responses that protect poultry populations against Newcastle disease by targeting these key viral components through live attenuated, inactivated, subunit, or recombinant vaccine formulations.

Peptide vaccines are a promising and innovative approach to immunization that utilizes short fragments of proteins, called peptides, as antigens to stimulate an immune response [22]. These vaccines offer several advantages, including specificity, safety, and ease of design and production [23]. Peptide vaccines can be tailored to target specific regions of a pathogen, such as a virus or cancer cell, minimizing the risk of unwanted side effects. They are particularly attractive for combating diseases like cancer, where traditional treatments may be less effective [24]. However, one challenge is that peptides alone may not always elicit a strong immune response, so adjuvants or other immune-boosting strategies are often used in combination with peptide antigens to enhance their effectiveness [25,26]. Overall, peptide vaccines represent a promising avenue in developing targeted and personalized immunotherapies for various diseases.

Computational techniques such as reverse vaccinology, immunoinformatics, and molecular docking play a crucial role in vaccine design to identify potential antigens and predict immune responses. These tools streamline the identification of vaccine candidates using pathogen genomes and target protein structures, saving time and resources compared to traditional methods. Molecular dynamic (MD) simulations are ideally suited to capture de novo the binding between receptor proteins and peptides [27] with an accuracy that can be similar to the one achieved for protein–ligand association. In vivo, analysis allows us to gain insight into some experimental conditions but does not explore mechanisms, while the MD could explore the molecular-level mechanisms of interactions [28,29,30].

In this study, we aim to design and evaluate a multi-epitopic peptide vaccine against NDV. We utilized molecular docking and MD simulations to assess binding affinity and stability, complemented by in vitro experiments to measure immunogenic responses.

## 2. Materials and Methods

### 2.1. Sequence Retrieval of F and HN Proteins and Antigenicity Prediction

The complete amino acid sequences of the F and HN proteins from indigenous Pakistani strains of Newcastle disease virus (NDV) were obtained from the National Center for Biotechnology Information (NCBI) in FASTA format. To assess the antigenicity of these proteins, we employed the VaxiJen v2.0 web prediction service to identify the potential antigenicity of the NDV proteins [31].

### 2.2. Consensus Sequence of F and HN Proteins

A consensus sequence was generated by performing multiple sequence alignment using the robust CLC Genomics Workbench software (version 24.0.1). This tool has been developed specifically for scientists and is designed to comprehensively analyze and visualize next-generation sequencing (NGS) data [32].

### 2.3. Epitope Prediction for CTL Epitopes

The Immune Epitope Database (IEDB) is a rich resource readily accessible to all researchers. It offers a wide range of epitope prediction and analysis tools, including well-validated and benchmarked methods for forecasting MHC class I and II binding. This study employed the Immune Epitope Database (IEDB) with its default threshold score to predict 9-mer cytotoxic T-lymphocyte (CTL) epitopes within the specified protein [33]. Furthermore, online servers, VaxiJen 2.0, AllerTOP 2.0 [34], and ToxinPred (version 2.0) [35], were used to screen the final CTL epitopes.

### 2.4. Epitope Prediction for HTL Epitopes

The 15-mer helper T lymphocyte (HTL) epitopes were predicted using the IEDB with the default settings within the selected protein. Furthermore, we screened these epitopes using VaxiJen v2.0, AllerTOP v2.0, and ToxinPred.

### 2.5. Prediction of B-Cell Epitopes

In the fight against viral infections, B-cell epitopes play a crucial role. These unique characteristics of B-cell epitopes can trigger antiviral responses. To identify B-cell lymphocytes (BCLs) for the selected protein, we employed the IEDB with a default threshold value. Additionally, we used VaxiJen v2.0, AllerTOP v2.0, and ToxinPred servers to select the final linear BCL epitopes.

### 2.6. Vaccine Construct and Its Properties

We developed several vaccine constructs by incorporating potential CTL, HTL, and BCL epitopes. Subsequently, we assessed each vaccine construct for allergenicity, antigenicity, and various physicochemical properties. To predict the allergenicity of each vaccine construct, we utilized the online server AllerTOP with a threshold value of 0.4. We employed VaxiJen 2.0 for antigenicity prediction, which accurately assessed our vaccine construct. To compute the physicochemical parameters of the vaccine, we used ProtParam, a tool that offers various metrics such as amino acid composition, protrusion index, half-life, aliphatic score, instability index, grand average of hydropathicity (GRAVY), and molecular weight. These metrics were computed using the ExPASY-ProtParam server [36].

### 2.7. Structure Prediction, Refinement, and Validation

We employed the Alphafold2.ipynb server to predict the 3D model of the primary vaccine construct [37]. Subsequently, we refined the vaccine model and assessed its overall quality by utilizing the Ramachandran plot [38]. The Ramachandran plot visually represents the distribution of torsion angles within a protein structure. It helps us to evaluate the allowed and disallowed regions of torsion angle values, which is crucial for assessing the quality of protein three-dimensional structures.

### 2.8. Docking

To study the interaction and binding affinity of the vaccine construct, we conducted docking simulations using the HADDOCK server [39]. Specifically, we docked the vaccine construct with two chicken receptors: TLR4 (PDB ID: 3mu3) and MHC1 (PDB ID: 3bew). HADDOCK is a widely used docking program known for its data-driven approach to docking, and it supports various types of experimental data, making it a valuable tool for this purpose.

### 2.9. MD Simulations

The MD simulations were performed for the peptides interacting with the MHC1 and TLR4 receptors using the Amber 2022 suite of programs [40]. The Amber ff19SB force field [41] was used for the receptors (chicken MHC1 and TLR4) and for the 40 aa peptide, which can be considered formally as a ligand. The two complex systems were immersed in a bath of TIP3P water molecules with a buffer of 14 Å, and counteranions (Na^+^ and Cl^−^) were added to neutralize the simulation boxes of respective dimensions 92.1 × 81.6 × 70.0 Å^3^ and 83.9 × 68.0 × 57.1 Å^3^. A reference system where only the peptide was simulated in the absence of any receptors was used to probe its intrinsic secondary structure (thus a reduced simulation box of dimensions 53.2 × 69.4 × 63.1 Å^3^).

The minimization of the three systems was performed using the steepest descent algorithm for 10,000 steps. Following this, the minimized structures were heated from 0 K to 300 K over 0.2 ns, with a time step of 1 fs. Each system then underwent equilibration for 1 ns at 1 atm pressure and 300 K, followed by a production run of 300 ns using a 2 fs time step with the SHAKE algorithm [42], with parameters set in the isothermal–isobaric NPT (constant temperature and constant pressure ensemble). The same computational protocols were applied to control simulations for the apo systems. Particle mesh Ewald and periodic boundary conditions were used in all simulations, using cut-off value of 12 Å for non-bonded molecular interactions. Convergence of trajectories was assessed using root mean square deviations (RMSDs) and visual inspection. Throughout the simulations, RMSD and root mean square fluctuations (RMSFs) were monitored using the bio3d R package [43]. Post-processing cluster analysis was conducted using cpptraj [44], employing a hierarchical algorithm based on RMS distance. For both systems, it was possible to identify one representative cluster with a weight higher than 65% (higher than 80% for TLR4 receptor).

#### 2.9.1. Principal Component Analysis

Principal component analysis (PCA) is a statistical technique used to analyze and simplify complex data sets. In this study, PCA was performed using the bio3d R package [43]. To capture the internal motion of the system, mass-weighted Cartesian coordinates were calculated. This approach produced a clustered representation of the data in terms of principal components (PCs), which are transformed variables. The trajectories of motion were primarily described by the first two PCs (PC1 and PC2). The resulting distribution pattern was consistent with a Boltzmann population [45].
G(X) = −kBT × ln P(X)(1)

In Equation (1), X stands for the response of the two principal components, kB denotes the Boltzmann constant, and P(X) describes the distribution of the system’s propensity along the first two principal components. The analysis of the first thirty eigenvectors revealed that the first three eigenvectors alone account for over 60% of the total variance for both receptors, a statistically significant finding that enhances the robustness of the subsequent analysis.

#### 2.9.2. Dynamic Cross-Correlation (DCCM)

The dynamic cross-correlation matrix (DCCM) was employed to analyze the time-dependent movements of Cα atoms, revealing information about the correlated and anti-correlated motions among the residues. This analysis of the Cα atoms of all residues was conducted using the specified equation for DCCM calculations as described in a previous study [45].

The MD simulation data were visualized using Gnuplot v5.4, a versatile command-line plotting tool compatible with multiple platforms [46], as well as MS Excel 2024. Trajectories were analyzed and depicted using the VMD software 1.9.4, which also generated cartoon representations. Protein–protein interactions were illustrated with LigPlot V2.2.4 [47] and Chimera 41.06.1311 [48].

### 2.10. Local and Peptide Vaccine Preparation and Synthesis

A mesogenic Newcastle disease virus strain, i.e., Mukteswar, for the current trial was taken from the viral vaccine section, CBP, VRI, Bacha Khan Chowk, KP, Peshawar. Embryo lethal dose-50 (ELD-50) for the viral harvest was calculated. The concentration of antigen had a minimum dosage of 10^9^ per mL in the aqueous phase.

The sequence of 40 amino acids as the final vaccine candidate was sent to GeneScript Biotech^®^ (Singapore) PTE Limited for synthesis. The synthesized peptide was received in lyophilized form.

The overall charge of the peptide was calculated to find whether the peptide is acidic or basic. The charge of the peptide as calculated was +2.96. The molecular weight of the peptide was 4440.29 g/mol, the isoelectric point was pH 9.78, the average hydrophilicity was -0.01, whereas the net charge at pH 7 was 2.96.

### 2.11. Reconstitution of Peptides

Three lyophilized vials were received, containing Peptide 1 (P1), Peptide 2 (P2), and Peptide 3 (P3), respectively, and immediately frozen till reconstitution. Ultrapure Millipore PCR water was used to reconstitute each peptide vial to a final concentration of 1 µg peptide/1 µL of water. Initially, 1/3 of the peptide from each of the vials was added separately to a 15 milliliter (mL) Falcon tube using a highly sensitive digital electronic scale (Ohauis Sartorius^®^, Parsippany, NJ, USA). Each peptide was separately added to water using a vortex mixer and was pooled together and homogenized by three 3 short pulses of 10 s each using an ultrasonic homogenizer (Comecta Ivymen^®^ system, Barcelona, Spain). The reconstituted peptides were kept at refrigeration temperature (2–8 °C) overnight and observed the next day for any precipitate formation. The next day it was observed that the peptides were homogenized properly and no precipitate was observed. The reconstituted peptides were poured into 1.5 mL cryovials at the rate (@) of 750 microliters (μL)/vial as stock solution. Fifteen (15) vials were prepared from 1/3 of the lyophilized peptide vials. These stock solutions along with the remaining lyophilized 2/3 of peptides were properly labeled, put in the rack, and stored at ultralow temperature (−80 °C).

Twenty percent (20%) adjuvant was prepared by adding 10 gm of alum (potassium aluminum sulfate (K_2_(SO_4_)·Al_2_(SO_4_)_3_·24H_2_O) or (KAl(SO_4_)_2_·12H_2_O)) in 50 mL of sterile phosphate buffer saline (PBS).

### 2.12. Preparation of Local and Peptide Vaccines

First, 50 ml of the local vaccine was prepared by mixing 25 mL of viral harvest (having a minimum ELD-50 = 10^9^ per mL) and 25 mL of alum adjuvant. The suspension was constantly stirred using a digital magnetic hot stirrer, model DRA-MS7-H550-Pro^®^, for 3 h in a biosafety cabinet, Type Class-II/A2, to ensure proper conjugation of antigen with adjuvant. The suspension prepared was centrifuged at 5000 rpm for 5 min and the supernatant was tested in a hemagglutination assay (HA).

Twenty-five micrograms (25 µg) of the synthetic peptide was mixed with four milligrams of potassium aluminum sulfate adjuvant. This was achieved by mixing 750 μL of peptide stock solution with 600 μL of 20% alum adjuvant. The suspension was properly mixed by a couple of short pulses of 10 s with a transducer of 1/8” diameter with a fixed frequency of 20 kilohertz (kHz) using an Ultrasonic Homogenizer^®^ Comecta Ivymen^®^ system, model CY-500, Spain to ensure proper capture of the peptide by the adjuvant. The suspensions prepared were stored at 2−8 °C for further use in the study.

### 2.13. Ethical Approval and Albino Mice

Ethical approval for the current study was given by the Veterinary Research Institute (VRI-23) and Institutional Research Ethical Board Committee (IREBC), No. KMU/IBMS/IRBE/7th meeting/2023/1209-28. An in vivo animal trial was performed following the Animal Research: Reporting of In Vivo Experiments (ARRIVE) guidelines [49] and NIH guide for the care and use of laboratory animals (NIH [50]). VRI-Peshawar Albino (VRI-PA) mice were granted by the laboratory animal house in charge, CPB, VRI, Bacha Khan Chowk, KP, Peshawar. A total of 63 8-week-old albino mice were received for this experiment.

### 2.14. Immunization of Albino Mice and Chicken with Local and Synthetic Peptide Vaccine

Three groups of mice each comprising twenty-one (21) mice were made and designated as Groups 1, 2, and 3. Group 1 was administered with 0.1 mL of the local ND vaccine (LNV). Group 2 was administered with 0.1 mL of synthetic peptide vaccine (SPV), whereas Group 3 was administered with 0.1 mL of PBS as a control group in separate cages for observation and sampling. The mice were restrained properly by the tail and neck to avoid biting and injected with a first shot (primer dose) of each vaccine at the dose rate (@) of 0.1 mL/sc near the region of the armpit (brachial plexus). A booster shot (booster dose) of each vaccine was injected on day 14 @ 0.1 mL/sc. Similarly, a control group was injected with 0.1 mL of sterile PBS.

One-hundred-day-old COBB-500 broiler chickens were purchased from Khali Sattar & Naushaba Khaleil (K&NNR^®^) Private Limited (Karachi, Pakistan). COBB-500 chickens have higher productivity and less growth during the fattening period compared with other cross broilers and is considered to be the most efficient. They were kept overnight in a control shed for brooding purposes. Flushing was performed with 1% sugar solution. Three groups of day-old chickens each comprising thirty-five (35) chicks were made and designated as Groups 1, 2, and 3. Group 1 was designated for LNV, Group 2 was designated for SPV, and Group 3 was designated as a control group. Briefly, the chickens were restrained delicately; Group 1 was administered with a standard dose of the first shot (primer dose) of LNV by an intraocular route on day 0, while the second shot (booster dose) of LNV was injected on day 14 by a subcutaneous route. Group 2 was administered 50 µg of SPV on day 0 and a booster dose on day 14 by a subcutaneous route. Similarly, the control group was administered with 0.1 mL of sterile PBS on day 0 and day 14. These groups were reared in separate compartments in an environmentally controlled shed at the VRI, Peshawar for recording of observations and sampling.

Blood samples were collected using a sterile, 1ml disposable syringe in a gel tube from mice and chicken groups on day 0, before vaccination, and days 14, 21, 28, 35, and day 42, after vaccination. The blood samples were stored in an incubator (37 °C) for 1 h. The serum was separated and stored at 4 °C. 

Serum samples were tested for antibody response using the hemagglutination inhibition (HI) test as described by [51]. The hemagglutination (HA) titer of the mesogenic genotype III ‘Mukteswar’ strain used in the study was 1:256, and 4 HA units of the antigen were used for the HI test.

## 3. Results

### 3.1. Antigenicity

Different genomic isolates (Appendix A) of local Pakistani NDV from the NCBI Virus database were retrieved. The VexiJen server analysis indicates that the HN and F proteins of NDV are highly antigenic compared to the other four proteins, with HN having an antigenicity score of 0.6 (threshold 0.4) and F having a score of 0.55. This highlights their potential antigenicity and relevance for vaccine development. The multiple sequence alignment analysis revealed that the identity percentages between our vaccine and common circulating genotype VII strains were approximately 85% for the HN protein and 90% for the F protein. These findings underscore the alignment of our vaccine design with prevalent NDV strains, enhancing its potential effectiveness. Understanding the immune response to these proteins is essential for further experimental studies to validate these predictions.

#### 3.1.1. Consensus Sequences of F and HN Proteins

Consensus sequences were developed by aligning multiple F and HN protein sequences from different isolates (Appendix A). The alignment process identified positions in the sequences that are conserved. The consensus sequence is valuable for understanding the fundamental characteristics of the protein, as it provides information about the most prevalent amino acids found in different isolates. This helps identify critical regions for targeting potential treatments, designing vaccines, or understanding how the protein functions across different strains or species. By focusing on conserved regions, researchers can develop interventions that are more likely to be effective against a broader range of isolates.

#### 3.1.2. CTL and HTL Epitopes

A total of 1021 CTL epitopes within the F protein with default threshold were predicted, of which 37 were antigenic, non-allergenic, and non-toxic. Similarly, 1101 CTL epitopes were predicted within the HN protein, of which 122 were antigenic, non-allergenic, and non-toxic. The selected epitopes of F and HN proteins are summarized in Table 1. A total of 1373 HTL epitopes within the F protein with default threshold were predicted, of which 541 epitopes were antigenic, non-allergenic, and non-toxic. Similarly, 448 HTL epitopes were predicted within the HN protein, of which 211 epitopes were antigenic, non-allergenic, and non-toxic. The selected and shortlisted HTL epitopes of F and HN proteins are summarized in Table 1.

#### 3.1.3. B Cell Epitopes

A total of 211 linear B-cell epitopes of F protein with a threshold of 0.4 were predicted, of which 51 epitopes were antigenic, non-allergenic, and non-toxic. Similarly, 41 linear B-cell epitopes of HN protein with a threshold of 0.4 were predicted, of which 13 were antigenic, non-allergenic, and non-toxic. However, only one epitope was selected for the final vaccine construct (Table 1).

#### 3.1.4. Final Vaccine Construct and Its Properties

Several vaccine constructs were designed; however, only one was selected based on the allergen, toxicity, and antigenicity scores. The final vaccine construct was 40 amino acids (aa) long, including F protein CTL epitope (9 aa), HTL epitope (15 aa), one HN protein HTL epitope (15 aa), and one F protein B-cell epitope (Table 1).

Appendix A presents the physicochemical properties of the vaccine construct. The multi-epitopic vaccine consists of 40 amino acids, with a molecular weight of 4441.27 Da and an isoelectric point (pI) of 9.9. It has an aliphatic index (AI) of 124.25 and an expected half-life of 5.5 h in mammalian reticulocytes (in vitro), more than 3 h in yeast (in vivo), and over 2 h in *E. coli* (in vivo). The grand average of hydropathicity (GRAVY) score is −0.347, with an instability index (II) of 33.71 and an antigenicity score of 0.7587. Additionally, vaccine construct 1 was determined to be non-allergenic and non-toxic.

The AlphaFold-predicted 3D protein structure is given in Appendix A. This model was found to be the best-refined model as compared to the other nine models. As revealed in the Ramachandran plot, the structural residues showed that 100.0% of residues were located in the most favored region, indicating that the vaccine’s overall quality was good (Appendix A). This model was used for molecular docking with the TLR4 receptor and MHC1 of chickens.

#### 3.1.5. Peptide Vaccine Docking and MD Simulations with Chicken TLR4 and MHC1

The top 1 HADDOCK results of the TLR4 receptor and peptide vaccine construct interactions are given in Appendix A. The first model having the highest HADDOCK score (−67.0 ± 1.9) was selected. The model was visualized using Ligplot+ (Version 2.2), shown in Appendix A. The docking of TLR4 and peptide vaccine included eight hydrogen bonding interactions. Four receptor residues interacted with a hydrogen bond (green lines), and other receptor residues interacted through hydrophobic interactions.

The best HADDOCK results of MHC1 receptor and vaccine construct interactions are given in Appendix A. The first model having the highest HADDOCK score (−73.3 ± 2.1) was selected. The model visualized by using Ligplot+ is given in Appendix A. A total of 12 interactions between MHC1 and the vaccine construct were observed. The eight receptor residues (MHC1) were interacting with the peptide vaccine through hydrogen bonds (green lines), and other receptor residues were interacting through hydrophobic interactions (red dashed lines).

MD simulations of the peptide interacting with either chicken MHC1 or TLR4 receptor complexes were carefully examined and compared. RMSD provides a first glimpse into the relative dynamic behavior. We first discuss the results of our 1 μs MD simulations for the chicken MHC1:peptide system, summarized in Figure 1. RMSD is stable after an initial opening between the two subparts of the receptor. After this initial motion, which does not impact the peptide binding, one can note that the complex exhibits a high yet stable RMSD value (16.2 ± 0.5 Å), which is a first prerequisite for an immunological response against NTV infection.

The RMSF values in Figure 1 reflect the flexibility of proteins in MD simulation, which is not very high near the peptide binding site and lies within the range of uniform flexibility but is higher than 4 Å for residues 180–271 forming the β2m domain. As expected, the RMSF is also enhanced in the peptide (residues 272–311) region, and the opening motion sketched in Figure 1 is also reflected in the evolution of the radius of gyration (Rg), which rapidly increases up to a stable value of 28.9 ± 0.4 Å, characterizing a large-amplitude motion.

To further assess the effect of protein folding on protein dynamics inferred from MD simulation, particularly its compactness, the radius of gyration (Rg) was monitored over the 1 μs MD trajectory (Figure 1). Limited fluctuations of Rg are essential for protein activity. The constant Rg centered around 28.9 ± 0.4 Å, after an initial increase corresponding to the opening in Figure 1. After this rearrangement, our visual inspection (and MM-GBSA post-analysis, given hereafter) indicates the firm binding and stability of the protein–peptide complex to elicit a stable immune response against NDV infection. The most flexible part corresponds to the β2m domain of the MHC1 receptor, far from the peptide binding.

The analysis of the 1 µs MD trajectory for the TLR4:peptide complex summarized in Figure 2 reflects a stable structure after the initial anchoring of the peptide. The RMSD graph of the peptide vaccine and TLR4 receptor is rapidly stable after this repositioning (16.2 ± 0.5 Å after 1 µs) until the end of the MD simulation period. We monitored the RMSF, which is relatively low (below 3 Å) for the TLR4 receptor and higher for the peptide—notably at the N- and C-termini. The Rg is stable and centered at 17.4 ± 0.2 Å, consistent with the representative cluster structures provided.

The cyan and violet colors correspond to positive and negative values, i.e., to correlated and anti-correlated motions. The correlated motions show the binding affinity of the peptide vaccine and MHC1 and TLR4 receptors. More correlated motions were detected in the peptide vaccine with TLR4 receptors (residues 1 to 45 and 100 to 144) as compared with MHC1. The is a small negative contribution from residues 130 to 150 and 180 to 250.

PC1 in Figure 3 accounts for the largest proportion of the variance (56.32%), followed by PC2 (17.95%) and PC3 (7.15%). This suggests that PC1 captures the most significant structural differences or motions in the system, likely reflecting key dynamic changes between peptide binding and the receptor. Similarly, the eigenvalue rank plot shows a steep drop after the first few components, suggesting most of the system’s variability, making them crucial for understanding conformational flexibility in the peptide–MHC1 interaction.

A large-scale protein dynamic can be computed using the essential dynamics. The motions of peptides and receptors are shown in Figure 3 and Figure 4.

Eigenvalue analysis (Figure 4) is a powerful mathematical tool showing the direction of motion and the stability of complexes used in protein dynamics to understand the vibrational, collective, and functional motions of proteins. It helps to reveal essential insights into the behavior and function of proteins in their dynamic environments.

The interaction profile highlights the predominance of weak, non-electrostatic interactions in the protein–peptide structure, primarily characterized by hydrophobic forces. Cluster analysis identifies a representative structure (population 81.7%) for the TLR4:peptide complexes extracted from a 1 µs (1000 ns) MD simulation and selected. The binding energy assessment using MM-GBSA, accounting for entropy corrections (normal mode analysis), reveals an overall binding energy of −15.7 ± 15.5 kcal·mol^−1^ for the peptide vaccine with TLR4 (Figure 5).

Similarly, the peptide vaccine exhibits a substantially stronger binding affinity with MHC1, with an overall binding energy of −36.8 ± 12.3 kcal·mol^−1^, also accounting for entropy corrections, underscoring its potential immunological significance (Figure 6). The representative structure shown in Figure 6 was obtained by cluster analysis (65.9% population).

#### 3.1.6. Comparative Immune Response of Mouse Groups to LNV and SPV

On day 0, Group 1 administered with the LNV showed a mean HI titer of 0.00, indicating no detectable immune response, while Group 2 injected with SPV had a mean HI titer of 4.00, suggesting some baseline immunity. Moving to day 7, both groups exhibited an increase in HI titers. The LNV group had a mean HI titer of 26.67 with a standard deviation of 7.64, while SPV had a slightly lower mean HI titer of 21.33 and a standard deviation of 8.93. By day 14, the LNV group demonstrated a substantial increase in the mean HI titer, reaching 106.67, with a relatively higher standard deviation of 37.93. The SPV group, on the other hand, showed a mean HI titer of 64.00, indicating a lower immune response compared to the LNV group, with a standard deviation of 0.00. On day 21, LNV group 1 maintained a steady mean HI titer of 128.00, while the SPV group showed a higher mean HI titer of 170.67, accompanied by a larger standard deviation of 63.54. Moving forward to day 28, the LNV group exhibited a mean HI titer of 85.33, with a standard deviation of 32.00, while the SPV group maintained the same mean HI titer of 170.67, but with an increased standard deviation of 67.45. The data suggest that both vaccines elicited an immune response over time, as indicated by the increasing HI titers as shown in Table 2 and Figure 7. However, no significant differences were observed between the two groups at any time point based on the calculated standard deviations and *t*-test analysis as shown in Table 3.

### 3.2. Comparative Immune Response of Chicken Groups to LNV and SPV

On day 0, Group 1 administered with LNV exhibited a mean HI titer of 14.4, suggesting a baseline immune response in some chickens. In contrast, Group 2 administered with SPV had a mean HI titer of 5.6, indicating lower baseline immunity. The standard deviation for LNV group at day 0 was 10.43, indicating some variability in the initial immune response, while the standard deviation for the SPV group was 2.19, indicating less variability. Moving to day 7, both groups showed an increase in HI titers. The LNV group had a mean HI titer of 28.8, with a standard deviation of 7.16, while the SPV group had a slightly lower mean HI titer of 25.6, accompanied by a standard deviation of 8.76. By day 14, Group 1 demonstrated a substantial increase in the mean HI titer, reaching 89.6, but with a relatively higher standard deviation of 35.05. The SPV group, on the other hand, showed a mean HI titer of 89.6, indicating a similar immune response to the LNV group, but with a similar standard deviation of 35.05. Progressing to day 21, the LNV group maintained a steady mean HI titer of 204.8, while the SPV group showed a higher mean HI titer of 204.8, accompanied by a larger standard deviation of 70.11. By day 28, the LNV group exhibited a mean HI titer of 102.4, with a standard deviation of 35.05, while the SPV group maintained the same mean HI titer of 204.8, but with an increased standard deviation of 70.11. The data suggest that both vaccines elicited an immune response over time, as indicated by the increasing HI titers as shown in Table 4 and Figure 8. However, no significant differences were observed between the two groups at any time point based on the calculated standard deviations and *t*-test analysis as shown in Table 5.

## 4. Discussion

Peptide vaccines offer a range of crucial advantages in the realm of immunology and vaccine development [52,53]. These vaccines are designed with precision to target specific segments of pathogens, such as viral proteins or antigens, resulting in a focused immune response against the most critical components. Their safety profile is notable, as they consist of short protein fragments, reducing the likelihood of adverse reactions compared to other vaccine types.

In vaccine designs, MD simulations play a pivotal role in the validation of peptide vaccines by providing a view of the interactions between the vaccine peptide and its target molecules within a biological system [54]. These simulations enable researchers to explore the vaccine peptide’s stability, flexibility, and binding affinity in a realistic physiological environment. By studying the complex and dynamic interplay of the peptide with immune receptors, such as MHC molecules or TLRs, MD simulations help assess the feasibility of inducing a robust immune response [55]. Additionally, MD simulations can identify potential conformational changes and interactions that may influence the vaccine’s efficacy, thus guiding the refinement and optimization of peptide vaccine candidates for enhanced immunogenicity and protective immunity [55,56].

In our study the analysis of the MD trajectory for the TLR4:peptide complex (Figure 2) reflects a stable structure after the initial anchoring of the peptide. The RMSD graph with the TLR4 receptor is stable, representing a good vaccine candidate. A stable interaction between a receptor and its bound peptide is essential for effective signal transduction and the initiation of immune responses [57]. The RMSD is an important property of proteins and ought to be stable for stable binding and interactions [58,59]. Stability of receptor–peptide complexes can provide valuable insights for the design of more potent vaccines and treatments for various infectious diseases [53,60,61,62,63]. This is a critical factor in modulating immunological responses [53,57,64].

This stability of our peptide:receptor complexes was confirmed through 1 µs MD simulations, with a more dynamic behavior for the peptide compared to the receptor. The two receptor–peptide complexes are thus stable here, and a closer analysis corroborates a stable binding site and stability at the microsecond range of the complexes. These outcomes agree with previous reports [53,57,64]. As shown in Figure 2, the Rg shows a stable compactness and folding of complexes, showing compactness and smooth folding properties which are also essential for proper function [65]. The compactness and folding stability of protein complexes are computationally assessed using Rg in MD simulations [66,67] which may lead to a strong immune response.

Similarly, the correlated motions in Figure 3 and Figure 4 show the good binding affinity of the peptide vaccine with MHC1 and TLR4 receptors. More correlated motions were detected in the peptide vaccine with TLR4 receptors. The DCCA is a computational technique employed in structural biology to delve into biomolecules’ correlated motions and dynamics. This is important to gain valuable insights into how various parts of a protein or other biomolecules move in respond to external stimuli [68]. By examining correlated motions within a protein’s structure, DCCA helps to decipher the dynamic mechanisms that enable proteins to perform their specific roles in various biological processes. PC1 in Figure 3 and Figure 4 accounts for the largest proportion of the variance (56.32%, 35.96%). This suggests that PC1 captures the most significant structural motions in the system, likely reflecting key dynamic changes between peptide binding and the receptor. Similarly, the eigenvalue rank plot suggests conformational flexibility in the pep-tide–MHC1 and –TLR4 interaction. The eigenvalue analysis is used in protein dynamics to understand proteins’ vibrational, collective, and functional motions. It is helpful to reveal essential insights into the behavior and function of proteins in their dynamic environments [69,70].

MM-GBSA per-residue decomposition of our peptide vaccine:receptor complexes demonstrated a good binding free energy. The MM-GBSA is a powerful computational technique used to dissect the energetic contributions of individual amino acid residues within a peptide vaccine when bound to a receptor such as MHC1 [71,72,73,74,75]. This analysis delves into the binding free energy, allowing pinpointing of the key interactions and contributions of specific residues in the peptide to the overall binding affinity. By identifying which residues make significant energetic contributions, MM-GBSA per-residue decomposition helps to rationalize peptide design, optimize binding, and gain a deeper understanding of the crucial interactions driving the peptide–MHC1 complex formation.

While the selected peptides exhibited promising characteristics in silico, the subsequent experimental validation phase was essential to confirm their immunogenicity and to evaluate their potential as vaccine candidates. The experimental validation of the chimeric multi-epitopic vaccine involved a comprehensive in vitro assay to assess its humoral response. In vitro experiments focused on measuring the vaccine’s ability to induce specific antibody responses in avian immune cells, which are vital for neutralizing viral particles and preventing viral entry into host cells [76]. The presence of specific antibodies against the F and HN proteins indicates the vaccine’s ability to trigger a targeted immune response against essential viral components. The antigenicity analysis revealed that the HN and F proteins exhibit scores of 0.6 and 0.55, respectively, indicating their strong potential for immunogenicity. Furthermore, the identity percentages between our vaccine and circulating genotype VII strains are approximately 85% for HN and 90% for F proteins. This close alignment reinforces the relevance of our vaccine design, suggesting that it may effectively stimulate an immune response against prevalent NDV strains [77,78].

To develop a novel candidate vaccine against NDV, we predicted and commercially synthesized a potentially novel candidate ND vaccine to be tested and compared with the existing ND vaccine using the Mukteswar strain. The Mukteswar strain, although a very famous candidate as a vaccine for provoking prophylactic immune response, causes mild symptoms of the disease with variable degrees of egg drop in layers [78,79,80]. The Mukteswar strain may not be used to provoke an immune response in day-old chicks and naïve birds, thus limiting the use of the Mukteswar strain in these cases. In our study, the mouse model could elicit a proper immune response like the immune response elicited in mice reported by [81]. The use of albino mice is a blessing to the field of biomedical research. Mouse models have been used for exploring fundamental pathophysiologic mechanisms [82], tumor detection, imaging, drug trial structure [83], human cancer [84], diabetic nephropathy [85], Parkinson’s disease drug development [86], and immunological responses against an infectious pathogen. Mouse models have provided breakthroughs in understanding the human immune system and have been extensively used for predicting various phenomena in humans because of their close similarity to humans [87,88,89,90,91]. The use of albino and transgenic mice has gained popularity in livestock research for the production of transgenic livestock [92], for studying human emotional chemosignals [93] and the role of ABCG2 in secretion into milk [94], and identification of iso-DON (deoxynivalinol) and iso-deepoxy-DON glucuronides as novel DON metabolites in mice and cows [95].

In the study, mice given the LNV had no immune response on day 0, while the SPV group showed some baseline immunity. However, on day 7, both groups showed increased HI titers, with LNV having a slightly higher response. On day 14, LNV showed a stronger immune response than SPV, but by day 21, SPV surpassed LNV in mean HI titer, which remained higher on day 28. This showed the importance of SPV in elucidating the immune response. Similarly, poultry birds, being the natural hosts for NDV, were also considered as an avian model in the current study for eliciting immune response. Baseline immunity in chickens can vary based on factors such as age, prior exposure to pathogens, and the specific vaccine type. Studies have shown that pre-existing immunity can affect the response to vaccination and should be considered when designing vaccination strategies [96,97]. In our study the group administered with the LNV exhibited no detectable immune response on day 0. This suggests that the chickens in this group had no pre-existing immunity to Newcastle disease. In contrast, the group injected with the SPV showed a mean HI titer of 4.00 on day 0, indicating some baseline immunity. This baseline immunity could be attributed to prior exposure or cross-reactivity with related antigens. The dynamics of immune responses to vaccines can vary based on the type of vaccine, the specific antigen(s) used, and the individual variability among birds. Rapid increases in antibody titers are common following vaccination, and variations in response can be influenced by factors like the quality of the vaccine, the vaccination protocol, and host factors [98,99]. Both groups showed an increase in HI titers from day 7 onwards, indicating that both vaccines stimulated an immune response against Newcastle disease. The LNV group had a substantial increase in mean HI titer on day 7 and day 21, maintaining a relatively steady titer. However, there was significant variability within this group, as indicated by the standard deviations (Table 4 and Table 5). The SPV group had a lower mean HI titer on day 7 compared to the LNV group but showed a rapid increase on day 21. However, the standard deviation for the SPV group was notably larger, suggesting greater variability in the immune response. Interestingly, the SPV group maintained the same mean HI titer on day 28 with an increased standard deviation, while the LNV group exhibited a lower mean HI titer. The SPV group exhibited a slower initial immune response compared to the LNV group, but it showed a significant rise in mean HI titers by day 21, suggesting its potential for inducing strong immunity over time [52]. Lack of significant differences between vaccine groups is not uncommon, especially in studies with a relatively small sample size. Vaccine trials can be influenced by individual variability in immune responses, and small sample sizes may not provide enough statistical power to detect significant differences [100,101]. Variability in vaccine response is a known challenge in vaccine development. It underscores the importance of conducting larger-scale studies to improve the statistical power and to evaluate the significance of observed differences [102,103,104]. In conclusion, both LNV and SPV against ND elicited immune responses in chickens over time. However, the variability within each group and the lack of significant differences between the groups suggest that further investigation with larger sample sizes and additional statistical analyses may be necessary to evaluate the significance of the observed differences in immune response.

In our study, we chose alum as an adjuvant for both LNV and SPV. Alum is one of the oldest adjuvants used in vaccine development approved by the Food and Drug Administration (FDA) [104,105,106]. Unlike emulsions, aluminum-based adjuvants interact with antigens through electrostatic forces in both aqueous and non-aqueous phases, as well as through hydrophobic interactions, van der Waals forces, and hydrogen bonds [107]. The first study to use alum as an adjuvant for NDV was conducted by [107]. Subsequent studies have successfully utilized various concentrations of alum, such as 50% alum gel for rabies virus absorption, 25% alum gel for Haemophilus paragallinarum, and 10%, 20%, 30%, and 40% alum for complete absorption of the ND virus. Based on previous findings [108,109], a concentration of 20% of alum gel was selected for the current study to reduce experimental costs and minimize the risk of tissue damage at the injection site. In our study, mice immunized with 20% alum gel showed an increasing antibody titer by day 21. These findings align with the results of a previous study [108] that showed a decreasing antibody titer on day 45 in chickens using 10% serum-grade alum. The immune response in chickens vaccinated with an ND vaccine conjugated with 20% alum gel adjuvant maintained consistent antibody titers from 3 weeks to 45 days post-inoculation. Yamanaka [110] reported that two weeks after vaccination with an inactivated ND vaccine containing Montanide, the immune response was more than eight times stronger than that elicited by an inactivated ND vaccine alone at the same time point. Stone et al. also observed similar consistent results. The previous studies conducted on chickens showed consistent results, either at day 14 or 21, but they differ from the findings of the current study in terms of the stability of the immune response [81,111,112].

The identity percentages between our vaccine and the common circulating genotype of NDV suggest that there is a high level of similarity, particularly in the F protein, however, some genetic variation exists between the vaccine strain and the circulating genotype VII strains. The slightly lower identity in the HN protein could suggest slight differences in antigenic properties, which may influence vaccine efficacy. However, the overall high identity, especially for the F protein, which plays a crucial role in viral fusion and entry, indicates that the vaccine should still provide a significant level of protection against the circulating NDV genotype.

## 5. Conclusions

The current study designed a multi-epitopic peptide vaccine targeting NDV and demonstrated its potential to elicit strong immune responses. The MD simulation revealed that the 40 amino acid peptide vaccine maintained robust binding stability within TLR4 and MHC1 receptor complexes. This stability was evidenced by consistent RMSD values and favorable binding energies. Both local and synthetic peptide vaccines showed significant immunogenicity in mice and chickens, as evidenced by increasing HI titers over time. The findings suggest that the multi-epitopic peptide vaccine is a promising candidate for effective NDV immunization.

## Figures and Tables

**Figure 1 vaccines-12-01250-f001:**
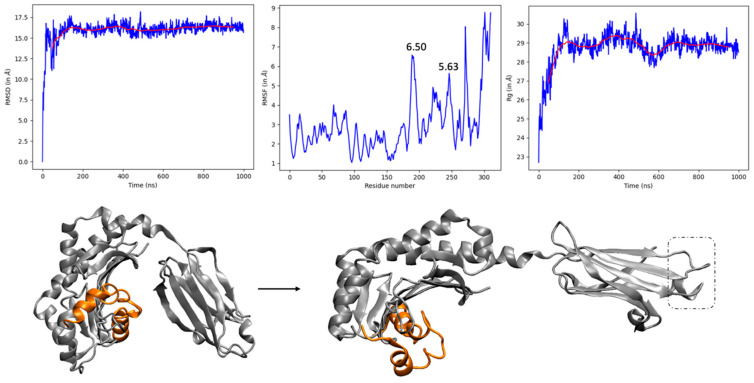
Time evolution of RMSD, RMSF, and Rg of peptide vaccine and MHC1 receptor complex over 1 μs. The RMSD graph of chicken MHC1 receptor interacting with the peptide vaccine becomes stable after an initial opening triggered by the very rapid disruption of salt bridge E31-R178. The peptide is shown in orange licorice mode, and the 188–195 loops are shown in the dashed box.

**Figure 2 vaccines-12-01250-f002:**
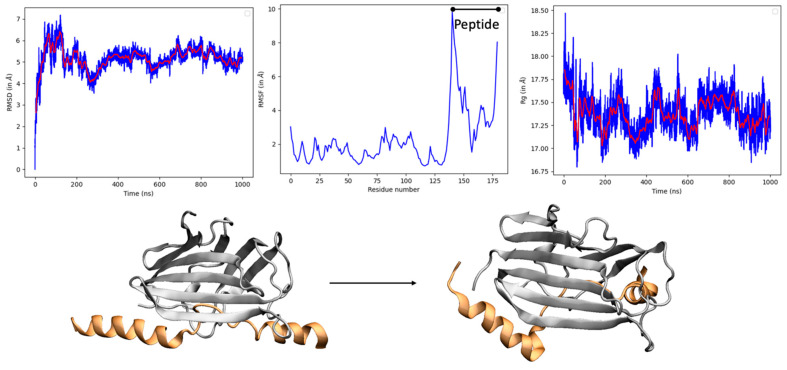
Time evolution of RMSD, RMSF, and Rg of peptide vaccine and TLR4 receptor complex over 1 µs.

**Figure 3 vaccines-12-01250-f003:**
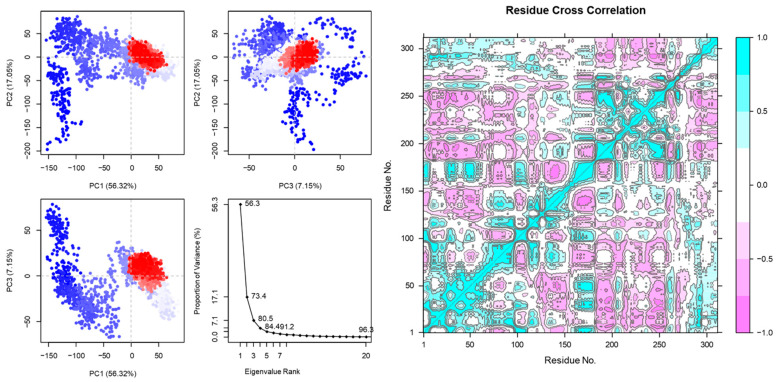
PCA (PC1, PC2, PC3) and DCCM of peptide vaccine and MHC1 receptors. PCA shows stable motion on both axes. In DCCM, violet indicates anti-correlated and cyan indicates correlated motion. The highest and lowest values have been normalized to −1.0 and 1.0, respectively. The direction of motion has been shown in the eigenvalue plot. The x- and y-axes denote the residue numbers of the peptide and MHC1 receptor complex. The color gradient illustrates the degree of correlation between the motions of residue pairs. Positive correlations are shown in blue. Negative correlations are shown in magenta. The residue numbers along the diagonal line are identical and indicate self-correlations.

**Figure 4 vaccines-12-01250-f004:**
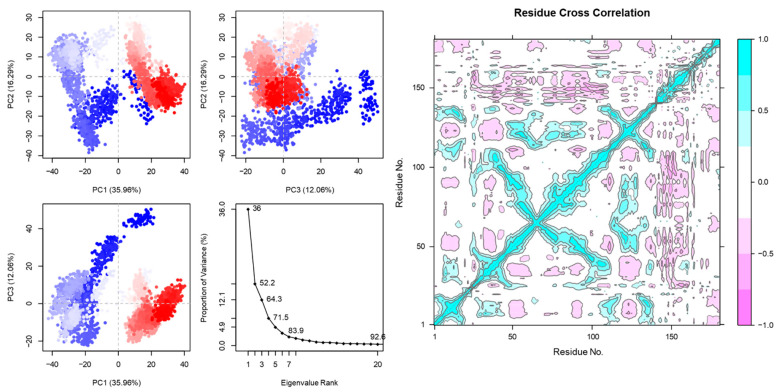
PCA (PC1, PC2, PC3) and DCCM of peptide vaccine and TLR4 receptors. PCA shows stable motion on both axes. In DCCM, violet indicates anti-correlated and cyan indicates correlated motion. The highest and lowest values have been normalized to −1.0 and 1.0, respectively. The direction of motion has been shown in the eigenvalue plot.

**Figure 5 vaccines-12-01250-f005:**
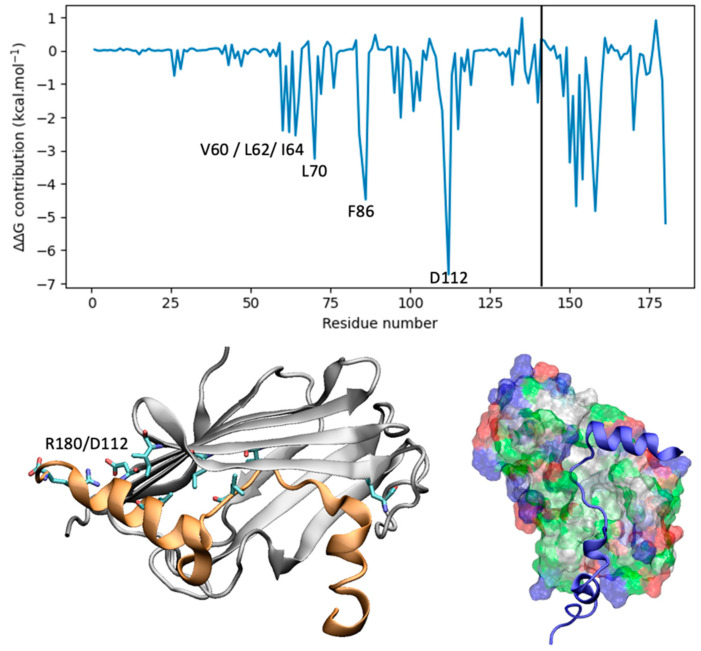
MM-GBSA per-residue decomposition for the peptide (residues 141–181) interacting with TLR4 receptor.

**Figure 6 vaccines-12-01250-f006:**
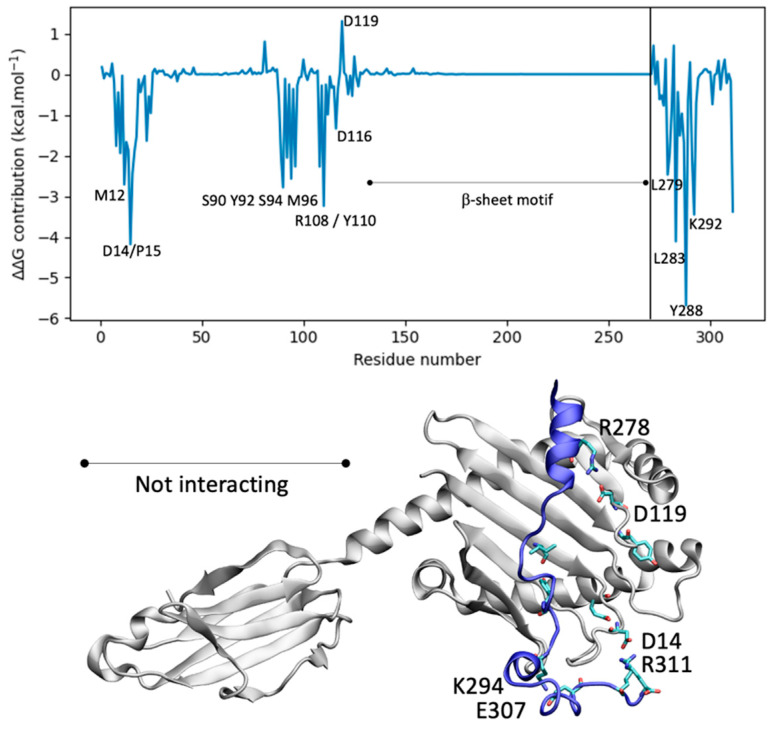
MM-GBSA per-residue decomposition for the peptide (residues 272–311) interacting with the MHC1 receptor.

**Figure 7 vaccines-12-01250-f007:**
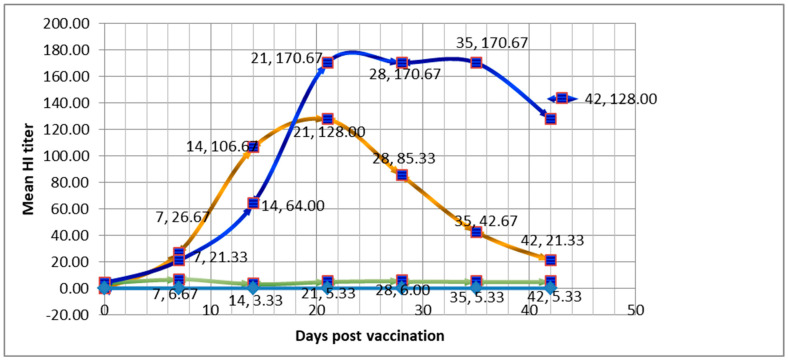
Comparative immune response of mice groups to local NDV vaccine (LNV = Brown line) and synthetic peptide vaccine (SPV = Blue line) along with a control group (Green line). Vertical axis of the figure indicates HI titer (From 0 to 200 indicates dilution from 1:0 to 1:200). Horizontal axis indicates days from day 0 to 50. Data Labels given with two figures at each mark, 1st figure indicates days and second figure indicates mean titer corresponding to the same day.

**Figure 8 vaccines-12-01250-f008:**
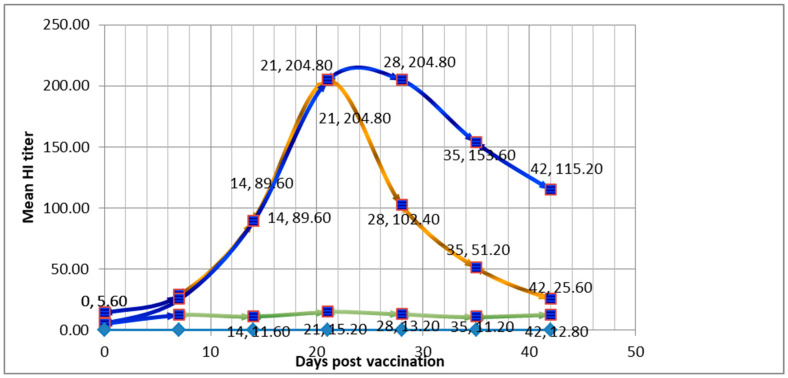
Comparative immune response of chicken groups to locally prepared vaccine (LPV = Brown line) and synthetic peptide vaccine (SPV = Blue line) along with a control group (Green line). Vertical axis of the figure indicates HI titer (From 0 to 200 indicates dilution from 1:0 to 1:200). Horizontal axis indicates days from day 0 to 50. Data Labels given with two figures at each mark, 1st figure indicates days and second figure indicates mean titer corresponding to the same day.

**Table 1 vaccines-12-01250-t001:** Selected HN and F protein epitopes and their properties.

Peptide	Length	Epitope	Protein	Allergenicity	Immunogenicity Score (>0.4)	Toxicity
Epitope_1	9	CTL	F	No	1.801	No
Epitope_2	15	HTL	F	No	0.702	No
Epitope_3	15	HTL	HN	No	1.7	No
Epitope_4	1	B cells	F	No	1.0	No

**Table 2 vaccines-12-01250-t002:** HI titer of the individual sample, mean, and standard deviation (SD) of serum samples obtained from albino mice in each group 0, 7, 14, 21, 28, 35, and 42 days after vaccination.

Day	Sample 1	Sample 2	Sample 3	Mean HI Titer	Mean ± Standard Deviation
Group 1 Administered a standard dose of alum-based LNV					
0	0	0	0	0.00	-
7	32	16	32	26.67	26.67 ± 7.64
14	128	128	64	106.67	106.67 ± 37.93
21	128	128	128	128.00	128.00 ± 0.00
28	64	128	64	85.33	85.33 ± 32.00
35	32	32	64	42.67	42.67 ± 15.97
42	16	16	32	21.33	21.33 ± 7.64
Group 2 Administered a 50 µg alum-adjuvanted SPV					
0	4	4	4	4.00	-
7	16	32	16	21.33	21.33 ± 8.93
14	64	64	64	64.00	64.00 ± 0.00
21	128	256	128	170.67	170.67 ± 63.54
28	256	128	128	170.67	170.67 ± 67.45
35	128	256	128	170.67	170.67 ± 63.54
42	128	128	128	128.00	128.00 ± 0.00
Group 3 Administered 100 µL of PBS as a control group					
0	4	4	4	4.00	4.00 ± 0.00
7	8	8	4	6.67	6.67 ± 1.25
14	4	2	4	3.33	3.33 ± 1.25
21	4	8	4	5.33	5.33 ± 2.31
28	2	8	8	6.00	6.00 ± 3.46
35	4	8	4	5.33	5.33 ± 1.15
42	8	4	4	5.33	5.33 ± 1.15

**Table 3 vaccines-12-01250-t003:** Comparison of HI titer, mean, and standard deviation (SD) of serum samples between albino mice in Group 1 and Group 2.

Day	Local NDV Vaccine (LNV) Group			Synthetic Peptide Vaccine (SPV) Group			*p*-Value
	Mean	SD	Sample Size	Mean	SD	Sample Size	
0	0.00	0.00	3	4.00	0.00	3	*p* = 0.195
7	26.67	7.64	3	21.33	8.93	3	
14	106.67	37.93	3	64.00	0.00	3	
21	128.00	0.00	3	170.67	63.54	3	
28	85.33	32.00	3	170.67	67.45	3	
35	42.67	15.97	3	170.67	63.54	3	
42	21.33	7.64	3	128.00	0.00	3	

The *p*-value was calculated using the independent *t*-test (two-tailed distribution with type-II error).

**Table 4 vaccines-12-01250-t004:** HI titer of the individual serum samples, mean, and standard deviation (SD) obtained from chickens in LNV and SPV and control groups at 0, 7, 14, 21, 28, 35, and 42 days after vaccination.

Day	Sample 1	Sample 2	Sample 3	Sample 4	Sample 5	Mean HI Titer	Mean ± Standard Deviation
Group 1 Administered a standard dose of alum-based LNV							
0	8	16	32	8	8	14.40	14.4 ± 10.43
7	32	16	32	32	32	28.80	28.8 ± 7.15
14	128	128	64	64	64	89.60	89.6 ± 35.05
21	128	128	256	256	256	204.80	204.8 ± 70.11
28	64	128	64	128	128	102.40	102.4 ± 35.05
35	32	32	64	64	64	51.20	51.2 ± 17.53
42	16	16	32	32	32	25.60	25.6 ± 8.76
Group 2 Administered a 50 µg alum-adjuvanted SPV							
0	4	4	4	8	8	5.60	5.6 ± 2.19
7	16	32	16	32	32	25.60	25.6 ± 8.76
14	64	64	64	128	128	89.60	89.6 ± 35.05
21	128	256	128	256	256	204.80	204.8 ± 70.10
28	256	128	128	256	256	204.80	204.8 ± 70.11
35	128	256	128	128	128	153.60	153.6 ± 57.24
42	128	128	128	128	64	115.20	115.2 ± 28.62
Group 3 Administered 100 µL of PBS as a control group							
0	4	4	4	8	8	5.60	5.6 ± 2.191
7	8	8	32	8	8	12.80	12.8 ± 10.733
14	4	2	4	32	16	11.60	11.6 ± 12.681
21	4	8	32	16	16	15.20	15.2 ± 10.733
28	2	8	8	32	16	13.20	13.2 ± 11.628
35	4	8	4	8	32	11.20	11.2 ± 11.798
42	8	4	4	32	16	12.80	12.8 ± 11.798

**Table 5 vaccines-12-01250-t005:** Comparison of HI titer, mean, and standard deviation (SD) of serum samples between chickens in LNV and SPV groups.

Day	Local ND Vaccine (LNV) Group			Synthetic Peptide Vaccine (SPV) Group			*p*-Value
	Mean	SD	Sample Size	Mean	SD	Sample Size	
0	14.40	10.43	5	5.60	2.19	5	0.32476772
7	28.80	7.16	5	25.60	8.76	5	
14	89.60	35.05	5	89.60	35.05	5	
21	204.80	70.11	5	204.80	70.11	5	
28	102.40	35.05	5	204.80	70.11	5	
35	51.20	17.53	5	153.60	57.24	5	
42	25.60	8.76	5	115.20	28.62	5	

The *p*-value was calculated using the independent *t*-test (two-tailed distribution with type-II error).

## Data Availability

Data are contained within the article and Appendix A are provided.

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
