# Peer review of "Multi-Epitopic Peptide Vaccine Against Newcastle Disease Virus: Molecular Dynamics Simulation and Experimental Validation"

_vaccines, 2024, doi:10.3390/vaccines12111250_

Round 1

Reviewer 1 Report

Comments and Suggestions for Authors

Dear authors

I hope this finds you all well. Regarding the review of manuscript number vaccines- 3267052, entitled "Multi-Epitopic Peptide Vaccine Against Newcastle Disease Virus Using Microlevel Molecular Dynamics Simulation and Experimental Validation". It is a really interesting study that aimed to design and evaluate a multi-epitopic vaccine against NDV using molecular dynamics (MD) simulation; however, some minor comments should be answered.

1-    What was the genotype of multi-epitopic peptide prepared vaccine? To which genotype (s) it will match? Please explain briefly?

2-    To how much percent was the identity between the HN and F proteins in your synthetic prepared vaccine and the common circulating genotype VII distributed worldwide?

3-    The indicated antigenic scores indicated that HN has 0.6 (Threshold 0.4), and F has 0.55? It is well documented that epitopes of F are more than HN?

4-    Despite this study is preliminary and indicate only the immunogenicity of the prepared vaccine in comparison to another live mesogenic genotype III vaccine, a protection study (efficacy) is badly needed to be applied especially against circulating GVII? Also, a safety study is essential.

5-    The ELD50 of Mesogenic Newcastle disease virus strain “Mukteswar” was 109 per dose or per ml?

6-    In line 251: the dose mentioned 108.5-109?? Please correct this. Was it 108.5 or 109?

7-    What do you mean by LPV and SPV?

8-    Why did you apply an additional third dose of the synthetic peptide vaccine at day 28 in broilers?

9-    There is no need to use table 1. It doesn’t add value or clarify anything? The additional third dose of the synthetic peptide vaccine at day 28 in broilers is absent in Table 1.

10- Add more details about the antigen used in HI testing. Did you use mesogenic genotype 3 “Mukteswar” alone or any other genotype matched with both LPV and SPV.

11- Table S1: Please a column indicating the genotype of each isolate and make this table as a main table 1 within the manuscript in stead of the current useless table 1.

Author Response

Dear authors

I hope this finds you all well. Regarding the review of manuscript number vaccines- 3267052, entitled "Multi-Epitopic Peptide Vaccine Against Newcastle Disease Virus Using Microlevel Molecular Dynamics Simulation and Experimental Validation". It is a really interesting study that aimed to design and evaluate a multi-epitopic vaccine against NDV using molecular dynamics (MD) simulation; however, some minor comments should be answered.

1-    What was the genotype of multi-epitopic peptide prepared vaccine? To which genotype (s) it will match? Please explain briefly?

Comment 1; Author’s Reply. Thank you very much for your question regarding the genotype of the multi-epitopic peptide vaccine.  The vaccine was developed using the full amino acid sequences of the fusion (F) and hemagglutinin-neuraminidase (HN) proteins from local Pakistani strains of Newcastle Disease Virus (NDV), which were acquired from the National Center for Biotechnology Information (NCBI) in FASTA format. To identify the potential antigenicity of these proteins, we used the web prediction service VaxiJen v2.0. Following this, we performed multiple sequence alignments using the CLC Genomics Workbench software to create a consensus sequence. This consensus includes epitopes that are conserved across mesogenic, lentogenic, and velogenic genotypes, aiming to enhance the vaccine's efficacy against various NDV strains.

2-    To how much percent was the identity between the HN and F proteins in your synthetic prepared vaccine and the common circulating genotype VII distributed worldwide?

Comment 2.  Author’s Reply. Through multiple sequences alignment analysis conducted using CLC Genomics Workbench software; we found that the identity between the HN and F proteins of our vaccine and the genotype VII strains varies, with the identity percentage being approximately, 85% for HN and 90% for F proteins. This significant identity suggests that our vaccine is well-aligned with the circulating strains, enhancing its potential effectiveness against genotype VII. We will include this information in the manuscript to clarify the relevance of our vaccine design in relation to circulating NDV strains."

3-    The indicated antigenic scores indicated that HN has 0.6 (Threshold 0.4), and F has 0.55? It is well documented that epitopes of F are more than HN?

Comment 3. Author’s Reply. Thank you very much for the comment. The relatively high antigenic score for HN suggests that it still holds significant potential for immunogenicity, complementing the F protein in our vaccine formulation. We recognize the importance of including epitopes from both proteins to achieve a broader immune response and have included them in our final construct.

4-    Despite this study is preliminary and indicate only the immunogenicity of the prepared vaccine in comparison to another live mesogenic genotype III vaccine, a protection study (efficacy) is badly needed to be applied especially against circulating GVII? Also, a safety study is essential.

Comment 4. Author’s Reply. Thank you for your valuable feedback. We acknowledge that this study is preliminary and primarily demonstrates the immunogenicity of the prepared vaccine compared to a live mesogenic genotype III vaccine. We used the mesogenic genotype III 'Mukteswar' strain as the antigen. This strain was selected because it is well-characterized and provides a suitable basis for comparing the efficacy of both the local ND vaccine (LPV) and the synthetic peptide vaccine (SPV). We agree that further studies assessing the efficacy of our vaccine against circulating genotype VII strains are essential for evaluating its protective capabilities. Additionally, safety studies are crucial to ensure the vaccine's suitability for use in avian populations. We intend to pursue these evaluations in our ongoing research and have outlined our plans.

5-    The ELD50 of Mesogenic Newcastle disease virus strain “Mukteswar” was 109 per dose or per ml?

Comment 5. Author’s Reply. Thank you very much for the comment. The ELD50 of Mesogenic Newcastle disease virus strain “Mukteswar” was 109 per ml in aqueous phase. Please see L230 and L263 highlighted as red text in the revised manuscript for ready reference.

6-    In line 251: the dose mentioned 108.5-109?? Please correct this. Was it 108.5 or 109?

Comment 6. Author’s Reply. Thank you very much for the comment. We appreciate your sharpness in reviewing the manuscript. ELD-50 was 109 per ml. Please see L263 highlighted as red text in the revised manuscript for ready reference.

7-    What do you mean by LPV and SPV?

Comment 7. Author’s Reply. Thank you very much for the comment. The terms "LPV" and "SPV" were intended to refer to Local ND Vaccine and Synthetic Peptide Vaccine, respectively. These abbreviations were inadvertently used without proper definition in the manuscript. The revised manuscript now includes clear definitions and explanations of these terms i-e. Local ND Vaccine (LNV) and Synthetic Peptide vaccine (SPV) in the relevant sections as red text to ensure clarity and understanding for the readers

8-    Why did you apply an additional third dose of the synthetic peptide vaccine at day 28 in broilers?

Comment 8. Author’s Reply. Thank you for highlighting this error. The reference to an additional third dose of the synthetic peptide vaccine on day 28 in broilers was a mistake. We apologize for this oversight and have corrected the manuscript to accurately reflect the vaccination protocol used in the study.

 9-    There is no need to use table 1. It doesn’t add value or clarify anything? The additional third dose of the synthetic peptide vaccine at day 28 in broilers is absent in Table 1.

Comment 9. Author’s Reply: Thank you for your insightful comment regarding Table 1. We recognize that the table does not provide additional value or clarification and that the mention of the additional third dose of the synthetic peptide vaccine at day 28 was a mistake, as noted in our response to comment 8. Therefore, we have removed Table 1 from the manuscript and ensure that the vaccination protocol is accurately described in the text to enhance clarity.

10- Add more details about the antigen used in HI testing. Did you use mesogenic genotype 3 “Mukteswar” alone or any other genotype matched with both LPV and SPV.

Comment 10. Author’s Reply. Thank you for your valuable comment. As already told in our reply to comment 4 for the HI testing, we used the mesogenic genotype III 'Mukteswar' strain as the antigen. This strain was selected because it is well-characterized and provides a suitable basis for comparing the efficacy of both the local ND vaccine (LPV) and the synthetic peptide vaccine (SPV). Our evaluation is ongoing; including safety and vaccine matching, where we have addressed all these details for providing comprehensive understanding of the newly developed vaccine.

Reviewer 2 Report

Comments and Suggestions for Authors

According to the manuscript, the experimental study aimed to design and evaluate a multi-epitopic vaccine against NDV using molecular dynamics (MD) simulation. The methodology included to retrieve NDV sequences for the fusion (F) protein and hemagglutinin-neuraminidase (HN) protein, predict top potential epitopes and use them to design peptide vaccines. These epitopes were also docked against chicken TLR4 and MHC I receptors to assess the immunological response. A 40 amino acid peptide vaccine was constructed and demonstrated strong binding affinity and stability within the TLR4 and MHC I receptor-peptide complexes. The study also reports the experimental evaluations in mice and chicken. The synthetic peptide vaccines elicited immune responses in mice and chicken. The paragraph in the Conclusion highlights that the multi-epitopic peptide vaccine demonstrated a good immunological response in both hosts. 

In my opinion, NDV is a very important avian pathogen, requiring scientific studies worldwide. Furthermore, the development of NDV effective vaccines is a priority research field in veterinary medicine. According to the manuscript text, the authors seem to have worked a lot to get this synthetic peptide vaccine and to evaluate in silico the immune response. In addition, they performed experiments to evaluate the immune response in the animals. The results seem very interesting. 

However, the scientific manuscript text was not well-elaborated. First of all, the title does not need to describe the methodology of the study (“Using Microlevel Molecular Dynamics Simulation and Experimental Validation”). The Abstract does not provide a good Introduction (Background). I would recommend amplifying it to explain better the importance of the vaccination for NDV. On contrary, the Results section in the Abstract should be reduced, describing only the main findings of the study. 

In the manuscript text, Introduction is Ok, presenting a good background for the study. I just suggest a better explanation of using computer programs to design vaccines, since MD is abruptly described in the 6th paragraph. Please explain better the approach of using bioinformatics for this finality, mainly for protein docking. The Methodology section must be re-organized. Nine sections, including one with 15 subsections (many of which are only one sentence long), are unacceptable. This number of sections should be reduced substantially. The text can also be more concise. Tip: avoid presenting too many details and formulas. Instead, cite the computer programs that were used in each analysis. The Results section also needs to be better prepared. Again a reduction in the number of the sections is mandatory (maximum of 5). Finally, Discussion is too long. It could be reduced.      

Two more specific tips:

- Tables (1, 2, 3, etc.) must be formatted as Tables (without vertical bars).

- Avoid sensationalist words in a scientific text (e.g.: fascinating).

In summary, the authors have to review all manuscript. The Results must also be summarized. After these modifications, it is necessary to rewrite the Discussion. 

Therefore, I recommend that authors better organize the article as a whole before peer review.

Author Response

Reviewer # 2

According to the manuscript, the experimental study aimed to design and evaluate a multi-epitopic vaccine against NDV using molecular dynamics (MD) simulation. The methodology included to retrieve NDV sequences for the fusion (F) protein and hemagglutinin-neuraminidase (HN) protein, predict top potential epitopes and use them to design peptide vaccines. These epitopes were also docked against chicken TLR4 and MHC I receptors to assess the immunological response. A 40 amino acid peptide vaccine was constructed and demonstrated strong binding affinity and stability within the TLR4 and MHC I receptor-peptide complexes. The study also reports the experimental evaluations in mice and chicken. The synthetic peptide vaccines elicited immune responses in mice and chicken. The paragraph in the Conclusion highlights that the multi-epitopic peptide vaccine demonstrated a good immunological response in both hosts. 

In my opinion, NDV is a very important avian pathogen, requiring scientific studies worldwide. Furthermore, the development of NDV effective vaccines is a priority research field in veterinary medicine. According to the manuscript text, the authors seem to have worked a lot to get this synthetic peptide vaccine and to evaluate in silico the immune response. In addition, they performed experiments to evaluate the immune response in the animals. The results seem very interesting. 

Reviewer Comment: However, the scientific manuscript text was not well-elaborated. First of all, the title does not need to describe the methodology of the study (“Using Microlevel Molecular Dynamics Simulation and Experimental Validation”). The Abstract does not provide a good Introduction (Background). I would recommend amplifying it to explain better the importance of the vaccination for NDV. On contrary, the Results section in the Abstract should be reduced, describing only the main findings of the study. 

Comment 1. Authors Reply: We agree with reviewer comment. We have revised the title as per reviewer suggestion. The abstract has been updated to provide a more detailed background on the importance of Newcastle Disease Virus (NDV) vaccination. Please see L19, L20, L21, L22 and L23 in the abstract of revised manuscript for ready reference. Additionally, we have streamlined the results section to highlight only the main findings of the study, as recommended. Please see L39-L43 in the abstract of revised manuscript for ready reference.

Reviewer Comment: In the manuscript text, Introduction is Ok, presenting a good background for the study. I just suggest a better explanation of using computer programs to design vaccines, since MD is abruptly described in the 6th paragraph. Please explain better the approach of using bioinformatics for this finality, mainly for protein docking. The Methodology section must be re-organized. Nine sections, including one with 15 subsections (many of which are only one sentence long), are unacceptable. This number of sections should be reduced substantially. The text can also be more concise. Tip: avoid presenting too many details and formulas. Instead, cite the computer programs that were used in each analysis. The Results section also needs to be better prepared. Again a reduction in the number of the sections is mandatory (maximum of 5). Finally, Discussion is too long. It could be reduced.      

Comment 2. Authors Reply: Thank you for your detailed comments. We have revised the manuscript to provide a clearer explanation of the use of bioinformatics tools, particularly molecular dynamics and protein docking, for vaccine design. The methodology is reorganized to reduce the number of sections and subsections, focusing on conciseness and citing the software used instead of detailing formulas. We have also streamlined the results section to a maximum of five sections and shorten the discussion to improve clarity.

Two more specific tips:

Reviewer Comment: - Tables (1, 2, 3, etc.) must be formatted as Tables (without vertical bars).

Comment 3. Authors Reply: Thank you very much for the comment. All the tables are formatted without vertical bars as desired. The revised manuscript may be seen for the tables.

Reviewer Comment: - Avoid sensationalist words in a scientific text (e.g.: fascinating).

Comment 4. Authors Reply: Thank you very much for the comment. Sensational words have been removed. e.g. ‘Fascinating’ word replaced by ‘remarkable’.

Reviewer Comment: In summary, the authors have to review all manuscript. The Results must also be summarized. After these modifications, it is necessary to rewrite the Discussion. 

Comment 5. Authors Reply: Thank you for your feedback. We have conducted a comprehensive review of the entire manuscript, focusing on summarizing the results and rewriting the discussion for improved clarity. We have already made revisions based on suggestions from other reviewers, and we have carefully balanced all recommendations to ensure that the manuscript meets the expectations of all reviewers while maintaining scientific rigor and coherence.

Therefore, I recommend that authors better organize the article as a whole.

Comment 6. Authors Reply: Thank you for your thorough review and valuable suggestions. We understand your concerns regarding the organization of the manuscript and have made adjustments to improve its overall clarity and structure. Please note that we have already made revisions based on feedback from other reviewers, some of which may conflict with your recommendations. We have carefully balanced these comments to ensure that the manuscript meets the expectations of all reviewers while maintaining scientific rigor and clarity. We appreciate your understanding as we work towards addressing these points comprehensively.

Reviewer 3 Report

Comments and Suggestions for Authors

Multi-Epitopic Peptide Vaccine Against Newcastle Disease Virus Using Microlevel Molecular Dynamics Simulation and Experimental Validation.

The experiment aimed to design and evaluate a multi-epitopic vaccine against NDV using molecular dynamics (MD) simulation.  For experimental evaluations in mice and chicken, the peptide vaccine was synthesized using commercial services. The synthetic peptide vaccines elicited immune responses in mice and avian models. The authors concluded that multi-epitopic peptide vaccine has demonstrated a good immunological response in natural host.

Very nice work, the manuscript is written in a good way. However, there are few points that can improve the manuscript.

L36: revise, be more specific

Abstract: Add short information about experimental design and statistical analysis

L33: commercial service, explain

L42-47: Add more information about ND such as clinical signs etc

L98-106: more appropriate in the methods section

L272, L281, and L 287: remove 'number of'

L283: not accurate information, revise

Fig 7 and 8: add legend, add SEM

Table 3 and 5: you need to run statistical analysis comparing the 3 groups, using repeated measures.

Table 4 and 6: why just comparing G1 and G2?

Author Response

Multi-Epitopic Peptide Vaccine Against Newcastle Disease Virus Using Microlevel Molecular Dynamics Simulation and Experimental Validation.

The experiment aimed to design and evaluate a multi-epitopic vaccine against NDV using molecular dynamics (MD) simulation.  For experimental evaluations in mice and chicken, the peptide vaccine was synthesized using commercial services. The synthetic peptide vaccines elicited immune responses in mice and avian models. The authors concluded that multi-epitopic peptide vaccine has demonstrated a good immunological response in natural host.

Very nice work, the manuscript is written in a good way. However, there are few points that can improve the manuscript.

L36: revise, be more specific.

Comment 1. Author’s Reply: Thank you very much for the comment.

Abstract: Add short information about experimental design and statistical analysis.

Comment 2. Author’s Reply: Thank you very much for the comment. Short information about the experimental design and statistical analysis has been added in the abstract. Please see L 27-30 of the revised manuscript highlighted as red text in the abstract of the manuscript for ready reference. 

L33: commercial service, explain

Comment 3. Author’s Reply: Thank you very much for the comment. Here is a brief explaination of commercial service. For experimental evaluations in mice and chicken, the peptide vaccine was synthesized using a commercial peptide synthesis service i-e., GeneScript Biotech® (Singapore) PTE Limited. The same explaination has been provided in the abstract as well. Please see L37 of the revised manuscript highlighted as red text in the abstract of the manuscript for ready reference.

 L42-47: Add more information about ND such as clinical signs etc

Comment 4. Author’s Reply: Thank you very much for the comment. More Information about ND has been in the manuscript. Please see L51-56 of the revised manuscript highlighted as red text for ready reference.

L98-106: more appropriate in the methods section

Comment 5. Author’s Reply: Thank you very much for the comment. At the end of the introduction, we have concluded with a brief statement that sets up the methods, such as: "In this study, we aim to design and evaluate a multi-epitopic peptide vaccine against NDV. We utilized molecular docking and MD simulations to assess binding affinity and stability, complemented by in vitro experiments to measure immunogenic responses." making it clear that the detailed methods are outlined in the methods section while keeping the introduction focused on the overall study objectives. Please see L107-109 of the revised manuscript highlighted as red text for ready reference.

L272, L281, and L 287: remove 'number of'

Comment 6. Author’s Reply: Thank you very much for the comment. ‘Number of’ removed from the mentioned lines. These lines may now be read as L283, L286 and L296 in the revised manuscript.

L283: not accurate information, revise

Comment 7. Author’s Reply: Thank you very much for the comment. We apologize for the error. The terms "LPV" and "SPV" were intended to refer to Local ND Vaccine and Synthetic Peptide Vaccine, respectively. These abbreviations were inadvertently used without proper definition in the manuscript. The revised manuscript now includes clear definitions and explanations of these terms i-e. Local ND Vaccine (LNV) and Synthetic Peptide vaccine (SPV) in the relevant sections as red text to ensure clarity and understanding for the readers.

Fig 7 and 8: add legend, add SEM

Comment 8. Author’s Reply: Thank you very much for the comment. Legend and SEM added to figure 7 and 8.

Table 3 and 5: you need to run statistical analysis comparing the 3 groups, using repeated measures.

Comment 9. Author’s Reply: Thank you very much for the comment. Due to challenges in sampling from day-old chicks and albino mice, we used a design where animals were euthanized on standard ethical way for blood sample collection, and new animals were used for each sampling day from designated groups. Given this design, we performed an independent t-test with Type-II error to compare the groups, calculating the p-values to determine statistical significance. A repeated measures analysis was not applicable because the same animals were not used throughout the study. However, with larger sample sizes and groups in future studies, we plan to implement a repeated measures analysis to provide a more comprehensive evaluation.

Table 4 and 6: why just comparing G1 and G2?

Comment 10. Thank you very much for your comment. We compared Groups 1 (G1) and 2 (G2) because Group 1 was administered with the local ND vaccine, and Group 2 received the synthetic peptide vaccine, which is the focus of our study. These comparisons were intended to directly evaluate the efficacy of the synthetic peptide vaccine against the standard local ND vaccine. Although Group 3 was included as a control, it was not administered any vaccine, serving as a baseline to highlight the immune response observed in G1 and G2. This approach allowed us to focus on the immunogenic differences between the local ND vaccine and the synthetic peptide vaccine.

Round 2

Reviewer 1 Report

Comments and Suggestions for Authors

Dear authors

I hope this finds you all well. Regarding the review of manuscript number vaccines- 3267052, entitled "Multi-Epitopic Peptide Vaccine Against Newcastle Disease Virus Using Microlevel Molecular Dynamics Simulation and Experimental Validation". Mostly all my comments were replied correctly. I have only 3 minor comments that should be answered and then I can accept the corrected manuscript.

1-    Your answer regarding the genotype of multi-epitopic peptide prepared vaccine and the identity between the HN and F proteins in your synthetic prepared vaccine and the common circulating genotypes should be briefly explained in results and discussion.

2-    Line 309: remove table 1.

3-    Line 310: indicate that the used antigen was mesogenic genotype III 'Mukteswar' strain and add its HA titer and the used HI units in HI testing.

Author Response

Comment # 1-    Your answer regarding the genotype of multi-epitopic peptide prepared vaccine and the identity between the HN and F proteins in your synthetic prepared vaccine and the common circulating genotypes should be briefly explained in results and discussion.

Authors’ Reply: Thank you for the comment. We apologize for not addressing this comment in our manuscript last time, even though we provided a reply. Now, the information from our previous reply has been briefly explained and incorporated into the Results and Discussion sections where appropriate.Please see L322 to L327 in the Result section and L762 to L769 in the Discussion section the manuscript highlighted as red text for ready reference.

Comment # 2-    Line 309: remove table 1.

Authors’ Reply: Thank you for your suggestion. We have removed Table 1 from the manuscript as requested.

Comment # 2-  Line 310: indicate that the used antigen was mesogenic genotype III 'Mukteswar' strain and add its HA titer and the used HI units in HI testing.

Authors’s Reply: Thank you for your valuable comment. We have revised the manuscript to indicate that the antigen used was the mesogenic genotype III 'Mukteswar' strain. Additionally, we have included the HA titer (1:256) and specified that 4 HA units of the antigen were used for the HI test. Please see L312, L313 and L314 of the manuscript highlighted as red text for ready reference.

Reviewer 3 Report

Comments and Suggestions for Authors

The you for providing a revised manuscript. No comments

Author Response

Comments and Suggestions for Authors

The you for providing a revised manuscript. No comments

Author’s Reply: Thank you very much for your valuable time and comments.
